# **Coupling Library Jcup3: Its philosophy and application**

Takashi Arakawa<sup>1</sup>, Takahiro Inoue<sup>1</sup>, Hisashi Yashiro<sup>2</sup>, and Masaki Satoh<sup>3</sup> <sup>1</sup>Research Organization for Information Science and Technology, Minato-ku, Tokyo, Japan <sup>2</sup>RIKEN Center for Computational Science, Kobe, Hyogo, Japan <sup>3</sup>Atmosphere and Ocean Research Institute, The University of Tokyo, Kashiwa, Chiba, Japan

**Correspondence:** Takashi Arakawa (arakawa@rist.jp)

**Abstract.** In this paper, we describe the design of the coupling library Jcup and report its various applications including the coupling between the global atmospheric and oceanic models with different grid systems. Jcup is a software library mainly focused on weather/climate models and was developed for the purpose of coupling components of various models. Jcup has flexibility in application to an unspecified number of components of earth system models. In order to achieve high order safety

and versatility, we divided the processes of a general coupling program into processes of changing and not changing the values of the data, and placed the former outside the program and under the control of the user. As a result, Jcup has two features:
1) that the correspondence relationship of grid indexes is used as input information, and 2) that the user can implement an arbitrary interpolation code. Jcup was applied to atmosphere–ocean coupling, IO component coupling, and seismic model–structure model coupling, and the validity and usefulness of the design were demonstrated.

#### 10 1 Introduction

Meteorological and climate models are constructed not only by the dynamic motion of the atmosphere but also by complex interactions of physical processes, such as radiation and clouds, or of various atmospheric boundaries, such as the ocean and land surface. After choosing what to include in the model among the elements constituting the phenomenon, weather/climate phenomena can be accurately reproduced and predicted using a simulation model. These elements should be modeled with

- the required accuracy to calculate the resolution and the integration period corresponding to the time-space scale of the phenomenon to be expressed. However, there is an upper limit to the performance of the computer executing the calculation. Consequently, there is a trade-off between the number of model components, the accuracy of the modeling of each component, and the time-space scale of the calculation. On the other hand, the modeling precision and the spatiotemporal scale are interrelated, because the phenomena to be expressed in the model are determined to some extent by the spatial scale. Alter-
- natively, the spatial scale of the calculation is determined according to the phenomena represented by the model. Therefore, the number of components that are modeled, and at what accuracy and to what degree of the spatiotemporal scale they are calculated, is adjusted and determined by the capacity of the computer obtained at that time and the content of the modeled phenomena. Although the performance improvement of the CPU alone slows down by increasing the degree of parallelization, the computing performance of the entire system and the memory capacity increase at a speed generally following Moore's
- Law. Accordingly, the spatiotemporal scale on which the model can be computed also improves, and, as a result, it becomes

possible and necessary to incorporate more elements into the system or to model each element more precisely. For example, according to the transition of the components of NCAR's CCSM (CESM) as stated by Washington et al. (Washington et al. (2009)), at the beginning of its development, the model comprised two components, namely the atmosphere (land surface) and the ocean. In the 1990s, sea ice and aerosol were added, and vegetation and the carbon cycle are currently being incorporated.

- Spatiotemporal scales can be improved relatively easily according to the performance of the computer. However, to improve the model's accuracy, a series of procedures is required, such as constructing equations suitable for the phenomenon to be expressed, building the program, executing the program, and confirming its validity. For this reason, work by experts and research institutes in this field is indispensable for incorporating new components into the system or elaborating each component. In summary, in the modern meteorological/climate model, individual component models developed by individual groups agency
- are executed at each spatiotemporal resolution in parallel. Such a structure is not only a natural analogy of weather and climate phenomena, but it is an appropriate mapping of the state of research communities involved in meteorological/climate research. In the structure of such a model, the components interact as is the case of expressed subjects, and it is necessary to exchange appropriate information on an appropriate spatiotemporal scale corresponding to each component when executing the model. Appropriate grid remapping is required according to the spatial scale of each component, but it is not preferable from the view-
- point of development efficiency and maintainability to separately develop and implement such a program for each component. For this reason, dedicated software responsible for coupling between components has been developed and used.

In this paper, software that executes such tasks is called coupling software or couplers. Generally, there are two types of coupling software, one of which is a program targeting a specific model. In this case, because target components are predetermined, it can be sufficient with coupling software specialized for a specific grid system, time scale, or coupling pattern. An example of

- this type of coupling software is the coupler used in NCAR CESM. The other type is coupling software developed for the general use. Since the specific target is not assumed, the structures of both the interface and program are determined depending on the extent of support on the grid system, the coupling pattern, and the interpolation method. As a representative of this type of coupler, an OASIS coupler has been developed mainly by CERFACS and is widely used in European meteorological research groups(Graig and Valcke (2017)). Scup was developed to target models of the Japanese Meteorological Agency and Meteo-
- rological Research Institute, but has a general-purpose interface that can be used for other models(Yoshimura and Yukimoto (2008)). Furthermore, MCT was developed as a basic software library for constructing a coupling program and cannot perform coupling alone. However, it can be thought of as a general-purpose coupling software (Larson et al. (2005), Jacob et al. (2005)). In addition to being used for CESM couplers, MCT is also utilized as a basic library for constructing the latest version of the OASIS coupler. These coupling software supports the existing grid system and interpolation method and provides coupled
- computing environments, but in order to deal with grid systems and interpolation calculations that software does not suppose, some kind of software modification is required. On the other hand, Jcup is a library developed to corresponds to various lattice systems and interpolation algorithms without modifing the program in the future and to enables coupled calculation of various patterns.

In this paper, we describe the design features of Jcup and the reasons for adopting such a design. We further clarify the usefulness of the design by cases where models were coupled. First, we explain Jcup's design, focusing on two features

in particular. Next, the reason for Jcup having adopted such a design is described from the following three aspects: 1) the characteristics of the weather/climate model as a simulation model, 2) the relationship between the research community and the development community, and 3) the essence of coupling the models. Finally, we clarify the usefulness of the design adopted by Jcup by three cases: atmosphere–ocean coupling, coupling of IO components, and coupling of earthquake structural models.

#### 5 2 Overview of Jcup

The execution pattern is implemented independently of the model components in some couplers, including the older version of the OASIS coupler (Valcke (2013)) or the JAEA coupler(Nagai et al. (2011)). However, under a massively parallel computing environment, which is mainstream today, it is not preferable in terms of computational efficiency to occupy a plurality of computation nodes by a coupling process with a relatively light computing load. Therefore, Jcup adopted an execution form that operates as a part of each model component.

Furthermore, there is generally a number of data exchange patterns: a parallel exchange in which the models use data from a preceding step of target models alternatively, and a serial exchange in which subsequent models continuously use data from preceding models. There is also a number of execution patterns for each component model: each component is executed in parallel as an independent binary, or a plurality of components is sequentially executed in one binary. As shown in Fig.1, Jcup

- supports a total of four patterns, namely two of data exchange and two of model execution. In addition, it is applicable to the case where these patterns are complexly combined by three or more component models as shown in Fig.2. There are three binaries, namely A, B, and C, in the figure. Component A is executed in binary A, components B, C, and D are executed in binary B, and component E is executed in binary C. In binary B, component B is executed in all MPI processes. Subsequently, component C is executed in certain processes, and component D is executed in the other process in parallel. The solid line in
- the figure indicates parallel exchange, and the dotted line indicates serial exchange. In reality, it seems that there is no case where such complicated execution and data exchange patterns are required, but Jcup is designed to be able to deal with such complicated cases. The interface related to data exchange and the data flow in the program are detailed in Arakawa et al. (2011). As described in the next section, Jcup is not completed as a coupling software, and to use it, procedures such as implementation of interpolation code by the user and the making of a mapping table are required. Therefore, Jcup is called a "coupling library"
- and not a "coupler." In this paper we use "coupling program" or "coupling software" as broad terms that include couplers and coupling libraries.

#### **3** Design philosophy

#### 3.1 Characteristics of a climate model as a target of coupled simulation

The objective of coupling software is to provide users with a software environment that enables them to couple multidisciplinary

simulation models and to perform coupled simulation. Jcup was developed for being used for various coupled calculations without limiting the field.

(a) parallel execution, parallel exchange

(b) serial execution, parallel exchange

(c) parallel execution, serial exchange

(d) serial execution, serial exchange

**Figure 1.** Coupling pattern:(a) and (b) show parallel data exchange, and (c) and (d) show sequential data exchange.(b) and (d) represent single-program coupling, and (a) and (c) represent multiprogram coupling.

For this purpose, the external conditions confronted by the coupling program can be summarized as follows:

- Each model has a grid structure suitable for the physical state expressed by the model. In addition, the optimum grid structure may change depending on external factors, such as computer architecture.
- For example, in the conventional global atmospheric model, latitude and longitude grids and spectral methods were used. However, to avoid an increase in calculation cost of Legendre transformation, models using grid structures that differ from those in conventional models, such as icosahedral grids, Yinyang grids(Baba et al. (2010)), and Cubic grid (Adcroft et al. (2004)), have recently been developed. Regarding the ocean model, grid point concentration in the polar region (Arctic Ocean) is a classical problem, and Stretch and tri-polar grids are therefore widely used. Furthermore, river models, which adopt an irregular grid system that expresses the catchment along the river channel(Yamazaki et al. (2014)) might become a coupling target.

5

 The interpolation method between models varies depending on the physical requirements of each model and cannot be uniquely determined.

In cases where the integration period is relatively short or when the system is not physically closed, it is unnecessary to strictly satisfy conservativity, and in such cases, simple linear interpolation might be sufficient. On the other hand,