# Peer review of "Coupling Library Jcup3: Its philosophy and application"

_Geoscientific Model Development, 2018_

## Referee Comment (RC1) · Anonymous Referee #1 · 6 Jul 2018

This publication attempts to document the Jcup coupling library used, for example, to couple the NICAM atmosphere with the COCO ocean. Unfortunately, this submission does not provide enough detail to adequately understand or evaluate the work. As the title suggests, the authors spend a lot of time on philosophy and explanation and not so much on implementation. There is quite a bit of superfluous background for a GMD audience, but very few implementation details. For example, they don't even mention what language or programming model is used, what algorithms are used for the library functionality or what choices they made in data structures - I got all that from browsing the source and not from the manuscript itself. While they provide some tables of a few interfaces, an architecture diagram might have been helpful to see how exactly a user would adopt this library and what functionality they could expect. There is a bit more

detail in the referenced 2011 paper (though still not enough even there) and this paper does not seem to add anything new beyond what was published there, other than some newer applications of the library.

In terms of advancing the field, most of the functionality reported here already exists in nearly all ESM coupling frameworks. For example, time representation appears to be using integer time intervals to avoid roundoff, similar to the more comprehensive Earth System Modeling Framework (ESMF) time manager that many groups use or have copied. The interpolation formulation uses a linear, static, sparse-matrix multiply (shown in 3 redundant code fragments in pp. 13-15) that is already used by essentially all other frameworks (e.g. MCT, OASIS, TEMPEST, ESMF). Much of the leading-edge work in this area is moving toward non-linear property-preserving remappings that utilize higher-order interpolation while enforcing monotonicity, vector properties (div, curl) as well as standard conservation constraints.

The authors make the claim that they have created a more general library, but as in most frameworks, the more general the functionality, the more burden is placed on the user. So in fact, the specific algorithms used to compute interpolation weights are left to the user as are other aspects of the coupling presented here. While again, more detail would allow a better judgement of this, it appears the cost of this generality is passed to the component model and this has more of a flavor of interface standard rather than a library. While not exhaustive, some browsing of the code appears to confirm that many of the interfaces are at a somewhat lower level of abstraction than is seen in many current couplers. At the same time, they have made some curiously restrictive assumptions like always moving data to the destination grid for remapping, where a more optimal choice would be to minimize data motion by performing calculations closer to the finer-resolution grid, whether that's the source or destination mesh.

The application section also lacked significant detail, including on what sort of architecture the the tests were run. The model sizes and mesh points/node also seemed to be in a very inefficient regime and well beyond a strong scaling limit in some cases. The

analysis was also somewhat inadequate. While the conclusions are probably correct, the timing profile was a bit too coarse to come to their conclusions definitively and they could have added additional timers to really isolate computational time and message latency vs. load imbalance (barrier time).

There are some additional minor nits here and there - like using lattice for mesh, conservativity rather than conservation, and CLF in place of CFL that are, I suspect, a result of non-native language translation.

I apologize if this review is very harsh. It would be great to see a proper documentation of the authors' approach in the literature. A paper that spent more time on some of the details of their library and a more thorough evaluation of their approach in real production configurations would be preferable over this submission.

---

## Referee Comment (RC2) · Anonymous Referee #2 · 15 Jul 2018

The study aims to give design details of new coupling library to couple different earth system model components. Unfortunately, the current status of manuscript makes it hard to judge its contribution and added value to the previous works related with model coupling when other existing coupling libraries such as OASIS, ESMF and MCT are considered. The most of the presented low-level details of the proposed library already exists in other coupling libraries and frameworks. For example, applying SMM (Sparse Matrix-Multiply) to interpolation weights is very common among model coupling libraries. The manuscript also lacks to have comparison with the current implementations and existing ones (OASIS, ESMF, MCT ...) in terms of scalability under different loads, flexibility and error margin different interpolation methods such as bilinear, conservative etc. using a set of analytical functions like ones provided by SCRIP

interpolation library developed by LANL.

Specific comments about individual sections can be found as follows,

Introduction: - need to mention other model coupling libraries such as ESMF, MCEL etc. to indicate the main difference of the current implementation than others. this is not clear in the current version of the manuscript. - figure 2 need to be redesigned. it is hard to understand the interaction between components. sequence diagram can be an option.

Section 3.4.3: - interpolation routines are not implemented as black box in the other open source coupling libraries such as MCT, ESMF, OASIS. user can always look at the code (i.e. subroutine that calculates interpolation weights) and make modifications if it is desired. for example in ESMF user can pass interpolation weights calculated outside of the library to handle interpolation between different computational grids. Or custom extrapolation methods can be implemented by using multiple routehandle. - it is better to use interpolation weights rather than mapping table in general. if mapping table is something different then interpolation weights, it must be explained detailed to prevent confussion.

Section 4.3: Fig. 3 must be revisited and simplified. it is very confusing and hard to understand.

Section 5.1: - need to give more detail about bit-to-bit reproducibility. how it is achieved. it is not only related with the coupling library. it is also related with MPI library itself expecially using reduction operator and limited floating point representation.

Section 5.2.1: - In Fig.5, please give detail of the labels used in x axis such as GL09RL01 in the caption of the figure or in the text. it seems that it is combination of rlevel and glevel but it is better to clarify it.

Typos: - in gereneral style for citation must be revisited such as (Graig and Valcke, (2017)) need to be fixed as (Graig and Valcke, 2017) - page 16, line 20: suggestion to

change "non-hydrostatic equation system" to "non-hydrostatic dynamical core" - page 20, line 14: CLF must be CFL

---

## Referee Comment (RC3) · Anonymous Referee #3 · 18 Jul 2018

The manuscript summarizes the state of the JCUP coupling software, which is tested with an atmosphere-ocean and an atmosphere-landsurface-ocean-rivers system. In the beginning of the manuscript some general remarks on the authors' oppinion on coupling are given. In the current state, the manuscript is not suitable for publication in GMD for two reasons: a) the general statements about coupling do not build on a complete review of the existing strategies, nor they provide new concepts, and b) the description of the JCUP coupler is more a technical report and does not describe new functionality to be adopted by other systems. The functionality of the JCUP coupler might be sufficient for the tested applications, but seems rather weak compared to existing systems like OASIS/MCT, ESMF, or YAC. It remains unclear, why the authors have not adopted an existing, more flexible system.

[Figure]

some more specific remarks:

P1L17: it is unclear, why the number of components contributes to the trade-off in performance.

P1L22: this is not true these days, since the model setup (incl. number of componentes) is mainly determined by the research question, not by the computer power

P2L2: double citation

P2L5: 1990s is a vaque statement, references are needed

P2L10: which analogy is seen by the authors? This section is rather philosophical.

P2L29: other coupling software like ESMF, YAC are missing - this is a weak review of the state of the art.

Section 3.1-3.3: The statements are not referenced and read as personal oppinions of the authors more than a description of their system or a review of the state of the art.

P6L17: reasearch communities do not use black-box tools, in my experience, most researchers build on open-source community tools, where it is possible to detect, trace and fix a bug.

P6, Section 3.3 is not well written and is not generic - the scheduler of the coupled system is missing. "appropriate timing" is not explained. It is also unclear, why the model timesteps have to be dependent on the exchange timing. (although it is obviously easier to implement the coupled system like this)

P6 Section 3.4: minimize number of bugs and response time to new grid requirements are not performance requirements of a coupling system

P7, Section 3.4.1: time control works as expected, this does not need to be mentioned here, this is not a technical report.

P8L5: the usefulness of a model coupler also lies in the possibility to calculate mapping

tables. This is a weak point, not a feature.

P8L18: in general, it cannot be assumed that the mapping can be calculated before the simulation - the interpolation weights have to be recalculated during the simulation for varying grids (e.g. vertical movement of layers in an ocean model).

———————————————

---

## Editor Comment (EC1) · S. Valcke (Editor) · 16 Aug 2018

Dear Authors,

I am sorry to write that we will not go further in the GMD discussion of your manuscript "Coupling Library Jcup3: Its philosophy and application".

Indeed, the 3 reviews are all very negative, noting the lack of implementation details about Jcup3 in your manuscript.

Of course, this does not mean that papers about couplers or coupling technologies are not welcome in GMD. In fact, there has been such papers published more or less recently in GMD. But to be suitable for publication, the manuscript should contain e.g. much more details on the implementation (how it works in practice, what is the language used, what are the functions supported and constraints, etc.), a section on applications using the coupler, and a section analysing the software performance.

So we invite you to rewrite your manuscript taking into account these remarks and possibly to resubmit it, once it will have been upgraded.

With best regards, Sophie Valcke

---

## Author Comment (AC1) · 4 Sep 2018

First of all, we appreciate the reviewers who carefully reviewed the article.

All reviewers criticized at first that there is no technically new aspect in the coupling library Jcup. As reviewers pointed out, the technically new aspects about coupling software are not mentioned in this paper.

The theme of this paper is to discuss that when the user community and the development community have a specific relationship, what kind of function should be had in the coupling library as the basic software, or what kind of function should not be had. The conclusion obtained from the consideration is that a coupling library as basic software should not include factors that impede extensibility/versatility, and as a result should

not have advanced functions that will require future renovation about it. The coupling library Jcup introduced in this paper is designed, implemented and utilized based on this policy.

In that sense, it is reasonable to point out that there are no technically new aspects, but, on the other hand, it is regrettable that the novelty or superiority of design concept which the authors most wanted to appeal was not appreciated. Of course, it is the responsibility of the authors that the discussion concentrated on technical novelty, not on design superiority, and we recognize that a considerable amount of rewriting is necessary for correct evaluation. Therefore, we withdraw the submission of the paper according to the judgment of the editor.

We appreciate once again to all the reviewers who gave valuable comments.
* * *